# The Role of Cellular Defense Systems of Ferroptosis in Parkinson’s Disease and Alzheimer’s Disease

**DOI:** 10.3390/ijms241814108

**Published:** 2023-09-14

**Authors:** Jie Chu, Jingwen Li, Lin Sun, Jianshe Wei

**Affiliations:** 1School of Physical Education, Henan University, Kaifeng 475004, China; jiec@henu.edu.cn (J.C.); lijw0107@163.com (J.L.); 2Institute for Brain Sciences Research, School of Life Sciences, Henan University, Kaifeng 475004, China; 3College of Chemistry and Molecular Sciences, Henan University, Kaifeng 475004, China

**Keywords:** ferroptosis, cellular defense systems, Parkinson’s disease, Alzheimer’s disease

## Abstract

Parkinson’s disease (PD) and Alzheimer’s disease (AD) are the most common rapidly developing neurodegenerative diseases that lead to serious health and socio-economic consequences. Ferroptosis is a non-apoptotic form of cell death; there is growing evidence to support the notion that ferroptosis is involved in a variety of pathophysiological contexts, and there is increasing interest in the role of ferroptosis in PD and AD. Simultaneously, cells may have evolved four defense systems to counteract the toxic effects of ferroptosis occasioned by lipid peroxidation. This review, which focuses on the analysis of ferroptosis in the PD and AD context, outlines four cellular defense systems against ferroptosis and how each of them is involved in PD and AD.

## 1. Introduction

Parkinson’s disease (PD) and Alzheimer’s disease (AD) are the most common rapidly developing neurodegenerative diseases [1,2,3,4]. Ferroptosis is a non-apoptotic form of cell death: iron-dependent regulated necrosis occasioned by uncontrolled lipid peroxidation-mediated oxidative damage to the cell membrane [5,6,7]. The exact mechanisms associated with ferroptosis are still being investigated; however, it is mostly thought that ferroptosis may be due to oxidation, a type of cell death of different cell types (including cancer cells, nerve cells, and cardiac cells) triggered by glutamate toxicity, which inhibits cystine uptake through the cystine/glutamate reverse transporter protein system xc—(xCT), ultimately leading to glutathione (GSH) depletion and oxidative stress [8,9,10]. Ferroptosis has received much scientific attention because of its unique morphological and biochemical characteristics and the forms of death that distinguish it from other cells [11] (Table 1). The cytological changes occasioned by ferroptosis are distinct from apoptosis and other forms of cell death and exhibit necrotic-like morphological changes, including reduced cell volume, loss of mitochondrial cristae, increased mitochondrial membrane density, and rupture of the outer mitochondrial membrane [10,12]. However, the nucleus retains its structural integrity, and there is no swelling of the cytoplasm or organelles, rupture of the plasma membrane, or formation of apoptotic vesicles [8,13,14].

Table 1 shows the main predisposing factor, morphological and biochemical features, and commonly detected indicators of ferroptosis, apoptosis, necroptosis, and autophagic cell death (modified from [9,15,16]).

Over the past decade, a large body of evidence has shown that ferroptosis plays an important role in a variety of diseases, including neurological disorders, cancer, metabolic disorders, and cardiovascular diseases [17,18,19]. Recently, there has also been increased focus on the role of ferroptosis in neurological disorders, with studies suggesting that ferroptosis is implicated in the pathogenesis of a variety of neurodegenerative disorders, including PD, AD, amyotrophic lateral sclerosis (ALS), and strokes, including acute ischemic stroke (AIS), spontaneous intracerebral hemorrhage (ICH), and subarachnoid hemorrhage (SAH). The ALS cell mutation conduction model downregulates the cystine/glutamate antiporter (SLC7A11) and glutathione peroxidase 4 (GPX4), leading to motor neuron ferroptosis, whereas the activation of nuclear factor erythroid 2-related factor 2 (NRF2) attenuates motor neuron ferroptosis and exerts neuroprotective effects in vitro and in vivo [20,21]. In a mouse model of PD, it has been demonstrated that the loss of GPX4 is responsible for the vulnerability and motor dysfunction of midbrain dopaminergic neurons [22]. In most eukaryotic organisms, ferritin is primarily recognized as an important intracellular iron storage protein, which is an essential component of iron homeostasis and is involved in a variety of physiological and pathological processes. However, in addition to being expressed in the cytoplasm, ferritin is found in the nucleus, mitochondria, and lysosomes. Extracellular ferritin is also found in serum, synovial fluid (SF), and cerebrospinal fluid (CSF) [23]. In neurological diseases, ferritin deposition has been shown to be associated with the pathophysiology of the disease, and ferritin deposition is one of the key factors leading to ferroptosis. Treatments to treat and prevent organ damage by inhibiting ferroptosis with drugs have received particular attention [12,24,25,26,27]. We focus on the study of ferroptosis in the context of PD and AD and provide an overview of four cellular defense systems against ferroptosis and how each of them is involved in PD and AD. Finally, with regard to cellular defense systems for ferroptosis, the challenges and future research directions are discussed.

## 2. Cellular Defense System for Ferroptosis

Iron is a crucial trace element involved in a variety of biological processes, including DNA synthesis, cellular respiration, and immune function, and is essential for the survival of living organisms [28]. In normal healthy cells, iron homeostasis is tightly regulated to balance systemic uptake, distribution, cellular uptake, storage, and export [29,30,31]. Imbalance in iron homeostasis is a key cause of ferroptosis, which is often accompanied by excessive cellular iron uptake and the release of intracytoplasmic iron, such as ferritin phagocytosis [19,32]. The core events of ferroptosis involve increased iron accumulation, impaired lipid repair systems, and lipid peroxidation, ultimately leading to membrane disruption and cell death [33,34]. Intracellular GSH depletion and reduced GPX4 activity, the inability of lipid peroxides to be metabolized by GPX4-catalyzed reduction reactions, and the generation of large amounts of reactive oxygen radicals (ROS) by Fe^2+^ via the Fenton reaction increase intracellular and lipid oxidative stress levels [35,36]. The lethal accumulation of lipid peroxides in cell membranes can eventually yield membrane damage and rupture, which leads to ferroptosis, damage to cell membrane structure’s, and severe effects on various organelle functions [37,38,39].

Accordingly, cells may have evolved four defense systems to counteract the toxic effects of ferroptosis induced by lipid peroxidation [40]. The most notable defense mechanism against ferroptosis is mediated by GPX4, which acts as a lipid peroxidase, reducing toxic lipid peroxides from cell membranes to nontoxic lipid alcohols, thereby alleviating ferroptosis [5,41,42]. In addition, ferroptosis suppressant protein 1 (FSP1) acts as a REDOX reductase to inhibit GPX4-independent ferroptosis by reducing coenzyme Q (CoQ) to dihydroubiquione (CoQH_2_) on the plasma membrane. Subsequently, CoQH_2_ acts as a lipophilic radical to trap antioxidants; thus, lipid peroxy radicals are detoxified [43,44]. GPX4 acts synergically with dihydrowhey dehydrogenase (DHODH) to convert CoQ to CoQH_2_; thus, lipid peroxidation and ferroptosis are inhibited [45,46]. Finally, GTP cyclohydrolase 1 (GCH1) mediates the synthesis of the powerful endogenous antioxidant tetrahydrobiopterin/dihydrobiopterin (BH_4_/BH_2_), which inhibits ferroptosis in a manner independent of GPX4 [47,48]. The specific regulatory mechanism is depicted in Figure 1.

### 2.1. GPX4-Mediated Cellular Defense System

GPX4 (i.e., glutathione-dependent peroxidase) is a member of the GPXS family. Glutathione peroxidase (GPX), which is a highly conserved evolutionary enzyme, utilizes reduced glutathione (GSH) as a cofactor to detoxify lipid peroxidation [10,41,42,50]. GSH, the main cofactor of GPX4, is a tripeptide composed of glutamate, cysteine, and glycine [34]. In contrast, the synthesis of GSH entails an amino acid reverse transporter protein (System Xc-) that mediates the uptake of cystine in exchange for intracellular glutamate and is a heterodimer consisting of two subunits, namely, SLC7A11 and SLC3A2 [51]. System Xc-, which is widely distributed in phospholipids, exchanges glutamate and cystine in a 1:1 ratio, with cystine being taken up into cells, where it is reduced to cysteine [52]. GSH is then utilized as a cofactor for GPX4, which acts as a lipid peroxidase, reducing lipid peroxides to lipid alcohols; thus, propagation of lipid peroxidation in the membrane is limited [34]. Therefore, GPX4 is one of the most crucial defense mechanisms for the cellular detoxification of lipid peroxides. GPX4 activity is essential for maintaining lipid homeostasis in cells, preventing the accumulation of toxic lipid ROS and maintaining redox homeostasis [53].

GPX4 knockdown may lead to substantial cell death and cell degeneration (e.g., hepatocytes and neurons), which can be alleviated by ferroptosis inhibitors. GPX4-regulated ferroptosis is involved in cancer development and progression [54], drug-resistant persistent cells are highly dependent on GPX4 for survival, and loss of GPX4 function leads to cellular ferroptosis and prevents drug resistance; thus, a novel avenue of cancer therapy is proposed [55]. GPX4-ablation-induced ferroptosis is responsible for cognitive impairment and neurodegeneration in neurodegenerative diseases and can be ameliorated by ferroptosis inhibitors [56]. Regarding tissue or cellular injury, the inhibition of ferroptosis by adaptive GPX4 upregulation provides protection against the various adverse factors that contribute to the injury [57,58]. Overall, GPX4 crucially facilitates the regulation of ferroptosis, affecting the development and progression of a wide range of diseases.

### 2.2. FSP1-Mediated Cellular Defense System

GPX4 is the main enzyme regulating ferroptosis; however, a number of cancer cell lines (e.g., non-small cell lung cancer PC9, melanoma A375, and Kuramochi ovarian cancer JCRB cells) are resistant to GPX4 inhibitors [44,55,59,60,61], and the sensitivity to ferroptosis induced by GPX4 inactivation varies considerably between cancer cell types, which indicates that a parallel ferroptosis defense system to GPX4 may exist [59]. In U-2 OS osteosarcoma cells treated with GPX4 inhibitor 1S, 3R-RSL3 (hereafter RSL3), a CRISPR/Cas9 screening was conducted using apoptosis and a cancer single-stranded RNA (sgRNA) subpool to identify FSP1 as a potent ferroptosis defense factor [43]. FSP1 expression was positively correlated with resistance to multiple GPX4 inhibitors (RSL3, ML210, and ML162), and the protective effect of FSP1 against cell death was specific to ferroptosis and not to cell death induced by cytotoxic compounds and/or pro-apoptotic conditions [44]. FSP1 functions as an NADH-dependent coenzyme Q10 (CoQ10) oxidoreductase; it converts CoQ in the cell membrane to CoQH_2_ [62,63], which acts as a lipophilic radical-trapping antioxidant that prevents ferroptosis from occurring and is unique in this role [64].

FSP1 was originally named flavoprotein apoptosis-inducing factor mitochondrial-associated 2 (AIFM2) because it is homologous to apoptosis-inducing factor (AIF or AIFM1), a mitochondrial pro-apoptotic protein [65]. However, FSP1 lacks the N-terminal mitochondrial targeting sequence in AIF, is not located in mitochondria, and does not promote apoptosis. Therefore, AIFM2 was renamed FSP1 [43,44]. Although the mechanism of regulation with FSP1 is not appropriately understood, the level of resistance to ferroptosis is positively correlated with the level of FSP1 expression in many cultured human cancer cell lines, and the overexpression of FSP1 protects cancer cells from ferroptosis both in vivo and in vitro [43]. FSP1 exists as an independent parallel system that can be utilized with GPX4 to inhibit phospholipid peroxidation and ferroptosis [66]. FSP1 is a molecular target that facilitates neural repair, and administration of phenylephrine treatment can inhibit ferroptosis in mice after cerebral hemorrhage by upregulating FSP1, thereby reducing white matter damage and promoting the recovery of motor and coordination abilities [67]. Experiments with human cartilage and mouse chondrocytes show that FSP1 treatment attenuates the development of osteoarthritis in GPX4 knockout mice [68]. Ginsenoside Rg1 alleviates sepsis-induced acute kidney injury, probably by inhibiting ferroptosis in tubular epithelial cells in the kidney via FSP1. Ginsenoside Rg1 reduced iron content, ferroptosis-related protein, and MDA levels, increased GPX4, FSP1, and GSH levels, and inhibited lipid peroxidation and ferroptosis responses. In addition, the inhibitory effect of ginsenoside Rg1 on ferroptosis response was counteracted by FSP1 knockdown. In cellular experiments, ginsenoside Rg1 increased the viability of renal tubular epithelial cells and reduced iron accumulation and lipid peroxidation during ferroptosis; its anti-ferroptosis activity was dependent on FSP1. Ginsenoside Rg1 alleviated sepsis-induced acute kidney injury, possibly by inhibiting ferroptosis in renal tubular epithelial cells in the kidney via FSP1 [69,70]. In conclusion, FSP1 plays different roles in various diseases and may be a good molecular target; the upregulation of FSP1 or the stabilization of FSP1 expression could provide a novel direction for the treatment of related diseases.

### 2.3. DHODH-Mediated Cellular Defense System

DHODH is an inner mitochondrial membrane enzyme, an enzyme essential for the de novo biosynthesis of pyrimidine-based nucleotides, which catalyzes the de novo synthesis of pyrimidine ribonucleotide and crucially affects the metabolism of cancer cells [71,72]. DHODH catalyzes the oxidation of dihydroorotic acid (DHO) to orotate, which is essential for the production of uridine-5′-phosphate (UMP), and DHODH inhibition leads to pyrimidine depletion; thus, the cell lacks the essential nucleotides it requires [73]. DHODH-CoQ10 and mitochondrial (mGPX4) act as local mitochondrial defense systems, inhibiting mitochondrial lipid peroxidation and preventing ferroptosis. DHODH couples with the reduction of CoQ10 to CoQ10H_2_ after the oxidation of DHO to whey acid [72]. CoQ10H_2_ acts as a radical-trapping antioxidant, reducing lipid ROS in the inner mitochondrial membrane. DHODH and mGPX4 form the two main defense arms to inhibit lipid peroxidation in the mitochondrial membrane, which must be disabled to promote mitochondrial lipid peroxidation and induce ferroptosis [74]. DHODH has been extensively analyzed as a potential target for cancer therapy due to the increased demand for nucleotides by rapidly proliferating cells [72]. DHODH exhibits a unique function in attenuating mitochondrial lipid peroxidation and ferroptosis, which is independent of its traditional role in the production of pyrimidine nucleotides. The inactivation of DHODH sensitizes GPX4-high cancer cells to ferroptosis inducers and enhances ferroptosis in GPX4-low cancer cells [75]. Mechanistically, DHODH acts in parallel with GPX4, independent of other pathways, to inhibit ferroptosis through the reduction of CoQ to CoQH_2_. Moreover, this study indicates that DHODH inhibitors selectively inhibit GPX4-high tumor growth by inducing ferroptosis [75]. Therefore, DHODH could be utilized as a new target for the inhibition of ferroptosis, and DHODH research provides novel ideas for the treatment of ferroptosis.

### 2.4. GCH1-Mediated Cellular Defense System

To identify alternative approaches to ferroptosis regulation, a set of novel ferroptosis suppressor genes was identified through a genome-wide activation library screen, which culminated in the identification of a novel GCH1-centred ferroptosis suppressor axis [47]. Tetrahydrobiopterin (BH_4_) is an active cofactor involved in redox processes and exhibits antioxidant properties both in vitro and in vivo [76]. Overexpression of GCH1, the synthetic rate-limiting enzyme of BH_4_, not only abolished lipid peroxidation but also exhibited strong protection against the two ferroptosis inducers (i.e., RSL3 and IKE) and against ferroptosis induced by the absence of GPX4, independently of other ferroptosis-related antioxidant systems [77,78,79]. Furthermore, GCH1 did not protect against apoptosis induced by apoptosis inducers, which indicated that GCH1 could act as a selective target for ferroptosis [80]. Elevated levels of CoQ10, a potent antioxidant that neutralizes lethal lipid peroxidation by trapping free radicals, were found in GCH1-overexpressing cells [47]. A subsequent metabolic analysis of GCH1 overexpression cells indicated that BH_4_ was responsible for the effective anti-ferroptosis effect of GCH1 overexpression and that the supplementation of GCH1 knockout cells with BH_2_/BH_4_ was sufficient to rescue the cells from ferroptosis; however, it was observed that this BH_4_ function did not considerably influence its protective effect against ferroptosis [77]. In conclusion, GCH1, which acts as a ferroptosis defense system, prevents the onset of ferroptosis and acts independently of known ferroptosis-pathway-related proteins.

## 3. Cellular Defense System for Ferroptosis in PD

Clinically, individuals with PD exhibit movement disorders such as bradykinesia and rigidity, as well as non-motor symptoms, including olfactory deficits, constipation, pain, anxiety, depression, psychosis, and cognitive impairment, which may develop into dementia [81,82]. The main pathological feature of the disease is the marked and progressive degeneration of dopaminergic neurons in the substantia nigra [83], which is associated with systemic progressive iron accumulation [84]; thus, dopamine depletion in the striatum, loss of neuromelanin, and the appearance of intracellular Lewy bodies composed of aggregated alpha-synuclein occurs [85,86]. PD is induced by the loss of neurons in several brain regions, particularly dopaminergic neurons in the substantia nigra, and is treated primarily by restoring dopamine levels in the brain [87]. Dysregulation of iron homeostasis and abnormal accumulation leading to the production of ROS is a feature of PD [88]. Numerous studies have shown that the amount of increased iron in PD nigrostriatal glial cells and dopaminergic neurons correlates with the severity of the disease [89]. Dysregulation of iron homeostasis mechanisms triggered by iron regulatory proteins, resulting in increased intracellular iron input or reduced output, may be responsible for iron accumulation in PD [27,90]. As research into the mechanisms of PD and ferroptosis continues, there is increasing evidence of a close link between the pathogenesis of PD and ferroptosis mechanisms [91,92]. Brain imaging studies have indicated that there is a link between iron deposition in the substantia nigra and the loss of dopaminergic neurons in PD patients, and that, with regard to PD, an imbalance in iron homeostasis crucially contributes to neuronal death [12,90,93,94]. Pathological studies have reported ferroptosis in 1-methyl-4-phenylpyridine (MPP+)-induced SH-SY5Y (human neuroblastoma) cells, and 6-hydroxydopamine (6-OHDA)-induced SH-SY5Y cells, where has been proven to cause pathophysiological symptoms of PD [95,96]. Furthermore, a similar phenomenon was observed in a 1-methyl-4-phenyl-1,2,3,6-tetrahydropyridine (MPTP)-induced PD mouse model [91]. In addition, studies have noted that the utilization of ferroptosis inhibitors can rescue dopaminergic neuronal death in PD [12]. A growing number of studies have indicated a close relationship between ferroptosis and PD [15,41], and cells may now have evolved three defense systems to counteract the toxicity of ferroptosis induced by lipid peroxidation and to provide neuroprotection for PD patients [97,98,99]. The specific regulatory mechanism is depicted in Figure 2.

### 3.1. Role of the GPX4-Mediated Defense System in PD

Decreased GPX4 in forebrain neurons and spinal cord motor neurons leads to a loss of spatial learning and memory function in mice and accelerates the development of motor neuron disease [56]. Ferritinase inhibitor and a diet rich in vitamin E prevent hippocampal neurodegeneration in the forebrain neurons of GPX4 conditional knockout mice [103]. Increased GPX4 after brain injury can inhibit further cellular damage induced by iron toxicity [104,105,106]. Therefore, the induction of GPX4 expression and activity may be a strategy to protect against neurodegenerative diseases in this context.

The current study, which examined GPX4 expression in post-mortem human brain tissue from individuals with and without PD, noted a significant reduction in overall GPX4 in the substantia nigra of PD patients [107]. Nigro-specific GPX4 knockouts induce lipid peroxidation and lead to typical PD without α-synaptic nucleoprotein oligomers, which indicates that iron poisoning occurs downstream of α-synaptic nucleoprotein oligomers and contributes to dopaminergic neuron loss. Furthermore, this study demonstrated that midbrain GPX4 overexpression ameliorates PD-related dyskinesia by alleviating oxidative damage and the loss of dopaminergic neurons. Therefore, the retention of GPX4 activity represents a potential strategy for early intervention in PD [22]. Dopaminergic neurons are the cofactors responsible for dopamine (DA) production, which exhibits unstable properties that can lead to oxidative toxicity in the iron-rich substantia nigra environment [108]. Using a model in which α-synuclein, DA, and iron act synergistically to exacerbate ferroptosis in the midbrain nucleus, which leads to a loss of dopaminergic neurons during PD, this study introduces a novel pathway in which iron-induced DA oxidation isolates GPX4 through the ubiquitin–proteasome system to degrade it, thereby disrupting GPX4 function and leading to lipid peroxidation, and, ultimately, ferroptosis [22]. In an MPP+-induced model of ferroptosis in SH-SY5Y cells, GPX4 expression was inhibited, and quercetin attenuated the change; thus, DA neurons were protected. The Nrf2 inhibitor ML385 inhibited the quercetin-induced increase in GPX4 protein expression, which indicated that the protective effect of quercetin is mediated by Nrf2 [109]. In summary, the GPX4-mediated ferroptosis defense system mitigates lipid peroxidation, protects DA neurons, and alleviates PD.

### 3.2. Role of the FSP1-Mediated Defense System in PD

FSP1 acts as a novel CoQ10 plasma membrane oxidoreductase (CoQ10 is a potent antioxidant) that prevents lipid peroxidation and protects cells from ferroptosis through the reduced form of CoQ10 [110]. In an animal model of MPTP-induced PD, FSP1 is reduced in the substantia nigra of MPTP mice and can be inhibited by apoferritin (an iron-free form of ferritin) pretreatment. The pathological feature of PD is the damage of dopamine neurons in the substantia nigra pars compacta. This study indicates that reduced FSP1 may lead to an oxidized CoQ10 state, which further exacerbates lipid peroxidation, leading to the development of ferroptosis and, ultimately, the degeneration of DA neurons in PD. However, apoferritin could effectively inhibit the production of lipid peroxidation through FSP1 upregulation, thereby effectively suppressing ferroptosis, preventing the loss of dopamine neurons in the substantia nigra pars compacta, and exerting its neuroprotective effect on PD [97]. We observed that probiotic strain lactococcus lactis MG1363-pMG36e-GLP-1 inhibited ferroptosis by promoting FSP1 expression, thereby reducing lipid peroxidation and exerting neuroprotective effects in PD mice [111]. In summary, the FSP1-mediated ferroptosis defense system can effectively inhibit ferroptosis; thus, the production of lipid peroxidation is inhibited, and its neuroprotective effects are exerted.

### 3.3. Role of the GCH1-Mediated Defense System in PD 

The GCH1 gene belongs to a family of GTP cyclizing hydrolases known to be involved in the biosynthesis of BH_4_ [112], a cofactor for tyrosine hydroxylase (TH), which is the rate-limiting enzyme for dopamine biosynthesis [113]. Thus, variants in the GCH1 gene may contribute to dopa-responsive dystonia [114]. Using high-throughput transcriptomics, the study observed significant correlations between PD-related genes and ferroptosis-related genes at the transcriptional level and protein–protein interactions, revealing 22 unique differentially expressed genes that may act as a link between ferroptosis and PD [115]. Notably, GCH1 is present in both PD-associated and ferroptosis-associated genes, and researchers have identified GCH1 as a PD risk gene in Chinese PD patients [116,117]. In summary, GCH1 mutations can affect the PD phenotype, and its defense system can exert a neuroprotective effect.

## 4. Cellular Defense System for Ferroptosis in AD

AD is the most common neurodegenerative disease characterized by progressive memory impairment and cognitive deficits associated with the formation of β-amyloid (Aβ) plaques and neuroprogenitor fiber tangles from abnormally phosphorylated tau protein (p-tau) [118]. Abnormal levels of iron-regulated molecules have been implicated in the pathogenesis of AD [119], and a number of inhibitors of ferroptosis, such as vitamin E and desferrioxamine, have shown clinical efficacy in patients with AD [27,120], effectively reducing neuronal death and memory deficits caused by ex vivo and in vivo Aβ aggregation [121]. Brain iron homeostasis dysregulation, xCT upregulation, and lipid peroxidation, all of which are pathological features of ferroptosis, are found in AD brain samples compared with normal brain samples [122,123,124,125]. In addition, in the symptomatic AD transgenic mouse model, free iron accumulates after Aβ formation in cortical regions of the brain, and the increase in iron leads to significant cognitive deficits [126,127,128]. In 5×FAD mice, cognitive impairment began to appear at 4 to 5 months of age, with severe impairment occurring at approximately 9 months of age. No lipid signatures of increased lipid peroxidation and ferroptosis were observed between pre-symptomatic 5×FAD mice (3 months old) and WT mice, and increased lipid peroxidation and ferroptosis were observed in symptomatic 5×FAD mice (9 months old) [128]. The histopathology of AD is characterized by extracellular aggregation of Aβ plaques and intracellular aggregation of neuroprogenitor fiber tangles [129]. Aβ_1–40_ (subtypes of Aβ) causes Aβ to cross the blood–brain barrier into the blood by inducing mitochondrial autophagy and inhibiting GPX4 from inducing pericytes to undergo ferroptosis, leading to the development of AD [130]. Magnetic resonance imaging (MRI) studies have explored the link between iron accumulation and Aβ and tau aggregation, confirming that iron dysregulation in neurons plays a crucial role in AD [131]. Ferroptosis, as a novel mechanism of AD pathophysiology, has attracted attention as a promising target for AD therapy [40,132]. Cells may now have evolved three defense systems to counteract the toxicity of ferroptosis and provide neuroprotection to AD patients. The specific regulatory mechanism is depicted in Figure 2.

### 4.1. Role of the GPX4-Mediated Defense System in AD

It was found that GPX4 levels were reduced in the brains of AD mice and that the inhibition of GPX4 decreased the sensitivity of the brain to the effects of ferroptosis during the pathogenesis of AD [13]. Decreased GPX4 resulted in neuronal loss in the hippocampal region of both neonatal and adult mice, which was accompanied by increased astrocyte activation, suggesting an inhibitory function of GPX4 in neurodegenerative disorders [133]. The results of this study show that the inhibitory function of GPX4 is not only a function of the brain but also a function of the brain’s ability to respond to the effects of ferroptosis. In symptomatic 5×FAD mice (9 months old) (5×FAD/GPX4) mice, increased expression of GPX4 inhibited neuronal loss and lipid ROS production in the frontal cortex while reducing Aβ formation in frontal cortical tissues and improving learning and memory abilities. The 5×FAD/GPX4 mice with reduced surface ferritin deposition markers exhibited reduced 4-hydroxynonenal (HNE) levels, suggesting that ferritin deposition is a key factor in the pathogenesis of AD [56,134].

It has been shown that tetrahydroxystilbene glycoside inhibits the associated desferrioxic anemia and ameliorates the pathogenesis of AD in APP/PS1 mice by activating the GSH/GPX4/ROS signaling pathway [135]. Furthermore, in male APP/PS1 mice, N2a cells exposed to Aβ42, erastin-stimulated HT22 cells (a cellular model of desferrioxic anemia), and LPS-induced BV2 cells in these models, forsythoside A attenuated AD pathology by inhibiting ferroptosis-response-mediated neuroinflammation via the Nrf2/GPX4 axis [136]. These findings suggest that GPX4-mediated inhibition of the defense system is associated with AD pathology.

### 4.2. Role of the FSP1-Mediated Defense System in AD

Upregulation of FSP1 is involved in the mechanism by which a ketogenic diet prevents chronic-sleep-deprivation-induced AD. A ketogenic diet prevents chronic-sleep-deprivation-induced cognitive deficits, Aβ deposition, and excessive p-tau protein. A ketogenic diet suppresses iron homeostasis dysregulation by downregulating the expression of transferrin receptor 1 (TFR1) and divalent metal ion transporter protein 1 (DMT1) and upregulating the expression of ferritin heavy chain 1 (FTH1) and iron transporter protein 1 (FPN1). A ketogenic diet promotes the xCT/GPX4 axis, elevates FSP1, and reduces malondialdehyde (MDA). A ketogenic diet prevents ferroptosis by promoting lipid peroxide removal through increased levels of the GPX4/xCT system and FSP1 [137].

### 4.3. Role of the GCH1-Mediated Defense System in AD 

A whole-genome sequencing study of AD in a Chinese population showed that GCH1 was a genetic risk locus for AD, and the association between GCH1 rs72713460 and changes in plasma matrix metalloproteinase-2 (MMP-2) levels suggested that, in addition to regulating neurotransmitter levels, GCH1 may play an important role in the immune system or in Aβ-related metabolic pathways [138].

## 5. Discussion

Cells prevent and improve PD and AD by counteracting the toxicity caused by lipid peroxidation due to ferroptosis through defense systems. The specific mechanisms are as follows: (1) a GPX4-mediated ferroptosis defense system can alleviate lipid peroxidation and protect DA neurons; (2) an FSP1-mediated ferroptosis defense system can effectively inhibit ferroptosis, thus inhibiting the production of lipid peroxidation and exerting its neuroprotective effect; (3) GCH1 mutation can affect the PD phenotype, and its defense system can play a neuroprotective role; and (4) GPX4-, FSP1-, and GCH1-mediated ferroptosis defense systems can effectively inhibit AD ferroptosis and play a neuroprotective role. 

Although a growing number of studies confirm that ferroptosis plays an important role in PD and AD, the deeper molecular mechanisms are unknown, and studies with clinical translatability are still being explored. The ferroptosis cell defense system may be a promising target for the treatment of PD and AD. As the discovery that ferroptosis is involved in the pathogenesis of neurological disorders, an increasing number of investigators have proposed targeting anti-ferroptosis to treat these diseases. An increasing number of emerging compounds targeting have been found to exert their therapeutic efficacy through the ferroptosis cell defense system. Compounds may exert to alleviate the PD syndrome by activating GPX4-mediated ferroptosis cellular defenses system [27], such as (-)-Clausen amide [139], β-Hydroxybutyric acid [140], Quercetin [141], Dl-3-n-butylphthalide [142], thonningianin A [143], paeoniflorin [144], α-Lipoic acid [145], deferoxamine [146], ferrostatin-1 [95], and idebenone [147]. Compounds such as apoferritin attenuate PD by modulating FSP1-mediated ferroptosis cellular defenses system [97]. Some compounds can attenuate AD symptom by regulating the GPX4-mediated ferroptosis cellular defenses system [27], such as salidroside [148,149], tetrahydroxy stilbene glycoside [135], forsythoside A [136], and γ-glutamylcysteine [150]. This provides more possibilities for discovering potential therapeutic agents and therapeutic targets for PD and AD, as well as helping to further explain the pathogenesis of PD and AD. The vast majority of existing studies have been conducted in animal or cellular models, with few clinical trials. Therefore, further studies in humans through randomized trial designs are necessary to draw definitive conclusions. Previous studies have indicated that most of the pathways associated with ferroptosis inhibition converge on the GPX4 pathway, and PD and AD are not an exception. The findings of FSP1 by Bersuker [43] and Doll [44] and others, which deconstructed the notion that GPX4 was the only pathway to inhibit ferroptosis, and the findings of Mao [75] and Kraft [47], which provided the field with two more endogenous inhibitors, namely, DHODH and GCH1, which act independently of GPX4 to protect the body from ferroptosis, have apparently revolutionized ferroptosis research.

However, research into these defenses is still mostly at the cancer stage, and the specific regulatory mechanisms are not effectively understood, particularly the DHODH pathway, which has not been analyzed in PD and AD research. Key scientific questions related to the ferroptosis cellular defense system remain unaddressed. These questions can be expressed as follows: How does the ferroptosis defense system affect physiological situations? How many ferroptosis cell defenses exist? How exactly do the defense systems function, and how are they balanced with each other? With respect to the role of the ferroptosis cell defense system in diseases such as cancer, future research is, therefore, crucial. It is proposed that the discovery and clinical application of the ferroptosis cell defense system will provide novel countermeasures and targets for the diagnosis and treatment of diseases such as PD and AD as research progresses.

## Figures and Tables

**Figure 1 ijms-24-14108-f001:**
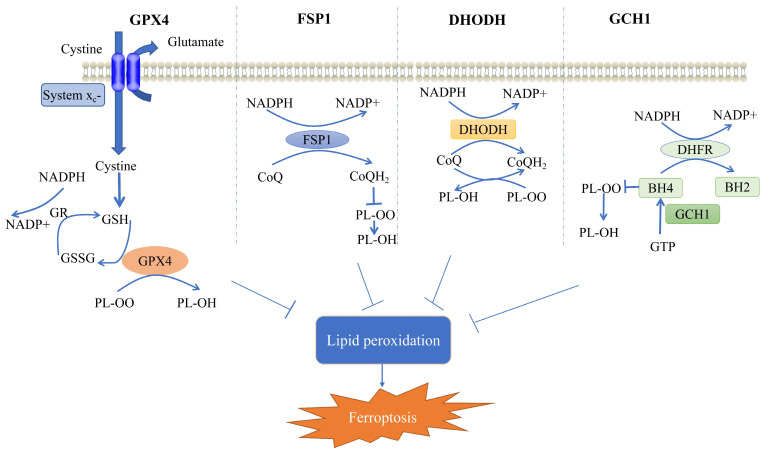
Mechanisms of ferroptosis cellular defense systems. Shows the four cellular defense systems of ferroptosis. GR, glutathione reductase; GSH, glutathione; GSSG, glutathione disulfide; GPX4, glutathione peroxidase 4; CoQH_2_, dihydroubiquione; CoQ, coenzyme Q; FSP1, ferroptosis suppressant protein 1; DHODH, dihydrowhey dehydrogenase; GCH1, GTP cyclohydrolase 1; DHFR, dihydrofolate reductase; BH_4_, tetrahydrobiopterin; BH_2_, dihydrobiopterin; PL-OO, phospholipid hydroperoxide; PL-OH, phospholipid alcohols; and NADPH, nicotinamide adenine dinucleotide phosphate. (Modified from [27,49]).

**Figure 2 ijms-24-14108-f002:**
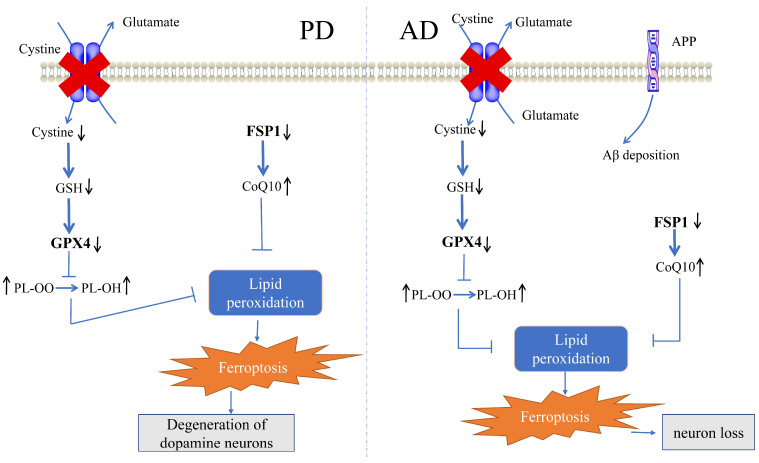
Mechanisms of ferroptosis cellular defense systems in PD and AD. GSH, glutathione; GPX4, glutathione peroxidase 4; CoQ10, coenzyme Q10; FSP1, ferroptosis suppressant protein 1; PL-OO, phospholipid hydroperoxide; PL-OH, phospholipid alcohols; and APP, amyloid precursor protein. (modified from [100,101,102]). Arrows indicate route directions.

**Table 1 ijms-24-14108-t001:** The main predisposing factor, morphological and biochemical features, and commonly detected indicators of ferroptosis, apoptosis, necroptosis, and autophagic cell death (modified from [9,15,16]).

Type	Predisposing Factor	Morphological Features	Biochemical Features	Commonly Detected Indicators
Ferroptosis	Accumulation of iron ions	Cell membrane: lack of rupture and blebbing of the plasma membrane, rounding-up of the cell; Cytoplasm: small mitochondria with condensed mitochondrial membrane densities, reduction or vanishing of mitochondrial crista, as well as outer mitochondrial membrane rupture; Nucleus: normal nuclear size and lack of chronmatin condensation	Iron and ROS accumulation; Activation of MAPKs; Inhibition of system Xc- with decreased cystine uptake GSH depletion and increased NAPDH oxidation; Release of arachidonic acid mediators; Δψm dissipation	Iron glutathione MDA GPX4 ROS LPO LDH cytotoxicity
Apoptosis	Gene regulation under normal physiological conditions	Cell membrane: plasma membrane blebbing, rounding-up of the cell; Cytoplasm: retraction of pseudopods, reduction of cellular volume; Nucleus: reduction of neclear volume, nuclear fragmentation, chromatin condensation	Activation of caspase; Oligonucleosomal DNA fragmentation; Δψm dissipation; PS exposure	Caspas series TUNEL Bcl-2 Bax
Necroptosis	Activated by the death receptor ligands and pattern recognition receptors of the innate immune system	Cell membrane: rupture of plasma membrane; Cytoplasm: cytoplasmic swelling (oncosis), swelling of cytoplasmic organelles; Nucleus: moderate chromatin condensation	Drop in ATP levels; Activation of RIPK1, RIPK3, and MLKL; Release of DAMPs; PARP1 hyperactivation	Hexosaminidase Calcein-AM Annexin-V ATP
Autophagic cell death	Nutritional deficiencies or hormone induction	Cell membrane: lack of change; Cytoplasm: accumulation of double-membraned autophagic vacuoles; Nucleus: lack of chromatin condensation	LC3-I to LC3-Ⅱ conversion; Substrate (e.g., p62) degradation	LC3 ATG series proteins

## Data Availability

Not applicable.

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
