# Peer review of "The Role of Cellular Defense Systems of Ferroptosis in Parkinson’s Disease and Alzheimer’s Disease"

_ijms, 2023, doi:10.3390/ijms241814108_

Round 1

Reviewer 1 Report

This review article highlights the involvement of ferropotosis in Parkinson’s disease (PD) and Alzheimer’s disease (AD), which recently draws much attention, and argues that ferroptosis could be a novel therapeutic target of PD and AD. 

This article dealing with the interesting issue must be valuable, but I evaluate that it should be written in a manner to attract broader attention. Since some of the compounds, proteins, mutant cells, and model mice described in this article are not necessarily familiar to all readers, I recommend adding brief explanation about their effects, functions, and characteristics. I suggest following words need additional explanation; ferritin, phenylephrine, ginosides, MPP+ and 6-OHDA, MPTP, desferritin, APP/PS1, 5xFD, 4-hydroxynonenal, tetrahydrosystilbene glycolside, coniferyl glycolside A, malondialdehyde, metalloproteinase-2.

I also feel that Figure 1 should be improved for the following reasons. In the GPX4 pathway, the relationship between GPX4 and GSH is not clearly depicted, and GSSH described in the legend is not on the figure. In the FSP1 and DHODH pathways, conversion of CoQH2 to CoQ10 looks as if it inhibits lipid peroxidation. In addition, the function of DHODH is not clearly depicted. In the GCH1 pathway, it is difficult to understand which component inhibits lipid peroxidation. 

Other comments

1) I wonder if iron or ferritin deposition is independent of altered ferroptosis defense systems in PD and AD. It might be better to mention it.

2) Line 237-238: 

I do not understand why alleviating damage and neuron loss enhance dyskinesia. 

Author Response

Response to Reviewer’s Comment

First of all, we would like to thank the reviewers for their invaluable suggestions and positive comments on the manuscript. We carefully examined these comments and have revised the manuscript point-by-point along the lines suggested.

The corrections in the revised manuscript are marked using red markers.

Reviewer 1

Point 1: I suggest following words need additional explanation; ferritin, phenylephrine, ginosides, MPP+ and 6-OHDA, MPTP, desferritin, APP/PS1, 5xFD, 4-hydroxynonenal, tetrahydrosystilbene glycolside, coniferyl glycolside A, malondialdehyde, metalloproteinase-2.

Response 1: Thank you very much for your suggestion. We have included a glossary at the end of the manuscript to clarify some of the terminology. The content is as follows:

Line 465-486 Glossary

Ferritin: ferritin is primarily recognized as an important intracellular iron storage protein, which is an essential component of iron homeostasis and is involved in a variety of physiological and pathological processes

Phenylephrine: a raise blood pressure drug

Ferroptosis inducers: a compound or treatment that can induce ferroptosis by boosting ferroptosis-promoting mechanisms and/or suppressing ferroptosis defense mechanisms

Ginsenoside Rg1: one of the active components of ginseng

MPP+: 1-methyl-4-phenylpyridine, it has been proven to cause pathophysiological symptoms of PD and has been widely used in the creation of PD models

6-OHDA: 6-hydroxydopamine, it is mainly used in the establishment of PD models

MPTP: 1-methyl-4-phenyl-1,2,3,6-tetrahydropyridine, it is a compound that causes selective degeneration of the substantia nigra after systemic administration and is used in modeling PD

APP/PS1: APP/PS1 mice, which are commonly used AD animal models

5xFD: 5xFAD mice, it is a widely used mouse model of AD

4-hydroxynonenal: HNE, quantitatively one of the most important products of lipid peroxidation

Forsythoside A: the main constituent of Forsythia suspensa

Matrix metalloproteinase-2: MMP-2, it can be involved in various intracellular mechanisms, including physiological and pathological processes, due to its proteolytic activity

Malondialdehyde: MDA, which is a secondary product of free radical lipid peroxidation.

Point 2: I also feel that Figure 1 should be improved for the following reasons. In the GPX4 pathway, the relationship between GPX4 and GSH is not clearly depicted, and GSSH described in the legend is not on the figure. In the FSP1 and DHODH pathways, conversion of CoQH2 to CoQ10 looks as if it inhibits lipid peroxidation. In addition, the function of DHODH is not clearly depicted. In the GCH1 pathway, it is difficult to understand which component inhibits lipid peroxidation.

Response 2: We modified Figure 1 as follows:

fig1

Figure 1. Mechanisms of ferroptosis cellular defense systems. Shows the four cellular defense systems of ferroptosis. GR, glutathione reductase; GSH, glutathione; GSSG, glutathione disulfide; GPX4, glutathione peroxidase 4; CoQH2, dihydroubiquione; CoQ, coenzyme Q; FSP1, ferroptosis suppressant protein 1; DHODH, dihydrowhey dehydrogenase; GCH1, GTP cyclohydrolase 1; DHFR, dihydrofolate reductase; BH4, tetrahydrobiopterin; BH2, dihydrobiopterin; PL-OO, phospholipid hydroperoxide; PL-OH, phospholipid alcohols; NADPH, nicotinamide adenine dinucleotide phosphate.

Point 3: I wonder if iron or ferritin deposition is independent of altered ferroptosis defense systems in PD and AD. It might be better to mention it.

Response 3: In the introductory section, we have adapted and added to the description of the problem. The content is as follows:

Line 50-64, The ALS cell mutation conduction model downregulates cystine/glutamate antiporter (SLC7A11) and glutathione peroxidase 4 (GPX4), leading to motor neuron ferroptosis, whereas the activation of nuclear factor erythroid 2-related factor 2 (NRF2) attenuates motor neuron ferroptosis and exerts neuroprotective effects in vitro and in vivo [20,21]. In a mouse model of PD, it has been demonstrated that the loss of GPX4 is responsible for the vulnerability and motor dysfunction of midbrain dopaminergic neurons [22]. In most eukaryotic organisms, ferritin is primarily recognized as an important intracellular iron storage protein, which is an essential component of iron homeostasis and is involved in a variety of physiological and pathological processes. However, in addition to being expressed in the cytoplasm, ferritin is found in the nucleus, mitochondria, and lysosomes. Extracellular ferritin is also found in serum, synovial fluid (SF), and cerebrospinal fluid (CSF) [23]. In neurological diseases, ferritin deposition has been shown to be associated with the pathophysiology of the disease, and ferritin deposition is one of the key factors leading to ferroptosis. Treatments to treat and prevent organ damage by inhibiting ferroptosis with drugs have received particular attention [12,24-27].

Point 4: Line 237-238: I do not understand why alleviating damage and neuron loss enhance dyskinesia.

Response 4: This issue was misrepresented in the manuscript, so the following changes have been made:

Line 273-275, Furthermore, this study demonstrated that midbrain GPX4 overexpression ameliorates PD-related dyskinesia by alleviating oxidative damage and the loss of dopaminergic neurons.

Reviewer 2 Report

In this manuscript, Chu et al. focus on ferroptosis and outlined the various defense mechanisms that the cells have evolved to counteract ferroptosis induced toxicity. The authors further discusse how these defense systems are involved in the context of neurodegenerative diseases including Alzheimer's disease and Parkinson's disease. The topic is interesting and relevant and the manuscript is well written. However, I have some comments and suggestions-

1. It would be informative to include a table/figure comparing ferroptosis to different other mechanisms of cell death

2. The figures should be referenced in the text.

3. In the introduction, the authors mention "The exact mechanisms associated with ferroptosis are still being investigated, but it is mostly thought that ferroptosis may be due to oxidation, a type of neuronal cell death triggered by glutamate toxicity, which inhibits cystine uptake through the cystine/glutamate reverse transporter protein system xc - (xCT), ultimately leading to glutathione (GSH) depletion and oxidative stress [8, 9]". Are other cells susceptible to ferroptosis? Is there a certain type of neuron that is mostly affected by ferroptosis? In neurological disorders, where does ferritin deposition take place? These details should be included.

4. The authors mention in several places about "cancer cell lines" (lines- 117, 135 etc.). It would be beneficial to include which particular cell lines were used and the appropriate references.

5.  The authors indicate in lines 120-121 - "A study that utilized a CRISPR/Cas9 screen identified FSP1 as a potent ferroptosis defence factor." What cellular system was used for the screen?

6.  "Ginsenosides inhibit ferritin precipitation in renal tubular epithelial cells via FSP1 and reduce acute kidney injury caused by sepsis [56, 57]." What is the exact mechanism of action?

7. At what age is ferroptosis observed in AD mice? Is it upsteram/downstream/coincidental with Abeta deposition? It will be beneficial to include these information in the text.

8. "In 5×FAD (5×FAD/GPX4) mice, increased expression of GPX4 inhibited neuronal loss and lipid ROS production in the frontal cortex, while reducing Aβ formation in frontal cortical tissues, and improving learning and memory abilities." At what age were these changes observed?

9. I would suggest that the authors include a table summarizing the various cellular defense mechanisms for ferroptosis and how these are compromised in AD and PD. It would also be valuable to discuss the current therapeutic advances (targeting ferroptosis) that have been made in the context of AD and PD.

The manuscript is well written and the english is easy to read. However, Some sentences are long and confusing and should be rephrased. For example Iines 44-49 "In addition, in these neurological disorders, ferritin deposition has been shown to correlate with the pathophysiology of the disease, and particular attention has been paid to therapeutic approaches for the treatment and prevention of organ damage through pharmacological inhibition of ferritin deposition, which has opened up new opportunities for the treatment of these disorders using pharmacological inhibition of ferritin deposition [11, 15-18]." 

Author Response

Response to Reviewer’s Comment

First of all, we would like to thank the reviewers for their invaluable suggestions and positive comments on the manuscript. We carefully examined these comments and have revised the manuscript point-by-point along the lines suggested.

The corrections in the revised manuscript are marked using red markers.

Reviewer 2

Point 1: It would be informative to include a table/figure comparing ferroptosis to different other mechanisms of cell death

Response 1: Thank you very much for your comments. We have added a table of comparing ferroptosis to different other mechanisms of cell death mechanisms to the manuscript. The content is as follows:

Line 39, Table 1 Shows the main predisposing factor, morphological, and biochemical, features, and commonly detected indicators of ferroptosis, apoptosis, necroptosis, and autophagy (modified from [9,15,16]).

table1

Point 2: The figures should be referenced in the text.

Response 2: We have included references in the figures in the manuscript.

Point 3: In the introduction, the authors mention "The exact mechanisms associated with ferroptosis are still being investigated, but it is mostly thought that ferroptosis may be due to oxidation, a type of neuronal cell death triggered by glutamate toxicity, which inhibits cystine uptake through the cystine/glutamate reverse transporter protein system xc - (xCT), ultimately leading to glutathione (GSH) depletion and oxidative stress [8, 9]". Are other cells susceptible to ferroptosis? Is there a certain type of neuron that is mostly affected by ferroptosis? In neurological disorders, where does ferritin deposition take place? These details should be included.

Response 3: Thank you very much for your advice. In the introduction we have made a detailed supplement. The content is as follows:

Line 27, “The exact mechanisms associated with ferroptosis are still being investigated; however, it is mostly thought that ferroptosis may be due to oxidation, a type of cell death of different cell types (including cancer cells, nerve cells, and cardiac cells) triggered by glutamate toxicity, which inhibits cystine uptake through the cystine/glutamate reverse transporter protein system xc - (xCT), ultimately leading to glutathione (GSH) depletion and oxidative stress.”

Current studies on ferroptosis on neurons have shown that motor neurons and dopaminergic neurons are affected by ferroptosis. Line 50, “The ALS cell mutation conduction model downregulates cystine/glutamate antiporter (SLC7A11) and glutathione peroxidase 4 (GPX4), leading to motor neuron ferroptosis, whereas the activation of nuclear factor erythroid 2-related factor 2 (NRF2) attenuates motor neuron ferroptosis and exerts neuroprotective effects in vitro and in vivo. In a mouse model of PD, it has been demonstrated that the loss of GPX4 is responsible for the vulnerability and motor dysfunction of midbrain dopaminergic neurons.”

Line 56, “In most eukaryotic organisms, ferritin is primarily recognized as an important intracellular iron storage protein, which is an essential component of iron homeostasis and is involved in a variety of physiological and pathological processes. However, in addition to being expressed in the cytoplasm, ferritin is found in the nucleus, mitochondria, and lysosomes. Extracellular ferritin is also found in serum, synovial fluid (SF), and cerebrospinal fluid (CSF). In neurological diseases, ferritin deposition has been shown to be associated with the pathophysiology of the disease, and ferritin deposition is one of the key factors leading to ferroptosis. Treatments to treat and prevent organ damage by inhibiting ferroptosis with drugs have received particular attention.” Therefore, we have included a description of ferritin deposition in this section.

Point 4: The authors mention in several places about "cancer cell lines" (lines- 117, 135 etc.). It would be beneficial to include which particular cell lines were used and the appropriate references.

Response 4: We describe this in "Cancer Cell lines" and cite relevant references. The content is as follows:

Line 137, “GPX4 is the main enzyme regulating ferroptosis; however, a number of cancer cell lines (e.g., non-small cell lung cancer PC9, melanoma A375, and Kuramochi ovarian cancer JCRB cells) are resistant to GPX4 inhibitors [44,55,59-61].”

Reference:

[44]       Doll S, Freitas FP, Shah R, Aldrovandi M, da Silva MC, Ingold I, et al. (2019). FSP1 is a glutathione-independent ferroptosis suppressor. Nature, 575:693-698.

[55]       Hangauer MJ, Viswanathan VS, Ryan MJ, Bole D, Eaton JK, Matov A, et al. (2017). Drug-tolerant persister cancer cells are vulnerable to GPX4 inhibition. Nature, 551:247-250.58.Zou Y, Palte MJ, Deik AA, Li H, Eaton JK, Wang W, et al. (2019). A GPX4-dependent cancer cell state underlies the clear-cell morphology and confers sensitivity to ferroptosis. Nat Commun, 10:1617.

[59]       Zou Y, Palte MJ, Deik AA, Li H, Eaton JK, Wang W, et al. (2019). A GPX4-dependent cancer cell state underlies the clear-cell morphology and confers sensitivity to ferroptosis. Nat Commun, 10:1617.

[60]       Viswanathan VS, Ryan MJ, Dhruv HD, Gill S, Eichhoff OM, Seashore-Ludlow B, et al. (2017). Dependency of a therapy-resistant state of cancer cells on a lipid peroxidase pathway. Nature, 547:453-457.

[61]     Tsoi J, Robert L, Paraiso K, Galvan C, Sheu KM, Lay J, et al. (2018). Multi-stage Differentiation Defines Melanoma Subtypes with Differential Vulnerability to Drug-Induced Iron-Dependent Oxidative Stress. Cancer Cell, 33:890-904 e895.

Point 5: The authors indicate in lines 120-121 - "A study that utilized a CRISPR/Cas9 screen identified FSP1 as a potent ferroptosis defence factor." What cellular system was used for the screen?

Response 5: We have supplemented this sentence in its entirety. The content is as follows:

Line 140, “In U-2 OS osteosarcoma cells treated with GPX4 inhibitor 1S, 3R-RSL3 (hereafter RSL3), a CRISPR/Cas9 screening was conducted using apoptosis and cancer single-stranded RNA (sgRNAs) subpool to identified FSP1 as a potent ferroptosis defense factor.”

Point 6: "Ginsenosides inhibit ferritin precipitation in renal tubular epithelial cells via FSP1 and reduce acute kidney injury caused by sepsis [56, 57]." What is the exact mechanism of action?

Response 6: This section is supplemented as follows:

Line 165, “Ginsenoside Rg1 alleviates sepsis-induced acute kidney injury, probably by inhibiting ferroptosis in tubular epithelial cells in the kidney via FSP1. Ginsenoside Rg1 reduced iron content, ferroptosis-related protein, and MDA levels, increased GPX4, FSP1, and GSH levels, and inhibited lipid peroxidation and ferroptosis responses. In addition, the inhibitory effect of ginsenoside Rg1 on ferroptosis response was counteracted by FSP1 knockdown. In cellular experiments, ginsenoside Rg1 increased the viability of renal tubular epithelial cells and reduced iron accumulation and lipid peroxidation during ferroptosis; its anti-ferroptosis activity was dependent on FSP1. Ginsenoside Rg1 alleviated sepsis-induced acute kidney injury, possibly by inhibiting ferroptosis in renal tubular epithelial cells in the kidney via FSP1.”

Point 7: At what age is ferroptosis observed in AD mice? Is it upsteram/downstream/coincidental with Abeta deposition? It will be beneficial to include these information in the text.

Response 7: Ferroptosis was observed in AD mice after the onset of symptoms, i.e., in mice of 9 months of age. Aβ can cause the occurrence of ferroptosis, so it is added in the manuscript.

Line 329, “In 5xFAD mice, cognitive impairment began to appear at 4 to 5 months of age, with severe impairment occurring at approximately 9 months of age. No lipid signatures of increased lipid peroxidation and ferroptosis were observed between pre-symptomatic 5xFAD mice (3 months old) and WT mice, and increased lipid peroxidation and ferroptosis were observed in symptomatic 5xFAD mice (9 months old) [128]. Aβ1-40 leads to an increase in Fe2+, an increase in lipid ROS, and a decrease in GSH-Px in pericytes, inducing ferroptosis by inhibiting GPx4 and xCT in pericytes, which can transport Aβ across the blood–brain barrier into the blood, leading to the development of AD [129].”

Reference:

[128]     Chen LJ, Dar NJ, Na R, McLane KD, Yoo KS, Han XL, et al. (2022). Enhanced defense against ferroptosis ameliorates cognitive impairment and reduces neurodegeneration in 5xFAD mice. Free Radical Biology and Medicine, 180:1-12.

[129]    Li J, Li M, Ge Y, Chen J, Ma J, Wang C, et al. (2022). β-amyloid protein induces mitophagy-dependent ferroptosis through the CD36/PINK/PARKIN pathway leading to blood-brain barrier destruction in Alzheimer's disease. Cell Biosci, 12:69.

Point 8: "In 5×FAD (5×FAD/GPX4) mice, increased expression of GPX4 inhibited neuronal loss and lipid ROS production in the frontal cortex, while reducing Aβ formation in frontal cortical tissues, and improving learning and memory abilities." At what age were these changes observed?

Response 8: This section is supplemented as follows:

Line 329, “In 5xFAD mice, cognitive impairment began to appear at 4 to 5 months of age, with severe impairment occurring at approximately 9 months of age. No lipid signatures of increased lipid peroxidation and ferroptosis were observed between pre-symptomatic 5xFAD mice (3 months old) and WT mice, and increased lipid peroxidation and ferroptosis were observed in symptomatic 5xFAD mice (9 months old).”

Point 9: I would suggest that the authors include a table summarizing the various cellular defense mechanisms for ferroptosis and how these are compromised in AD and PD. It would also be valuable to discuss the current therapeutic advances (targeting ferroptosis) that have been made in the context of AD and PD.

Response 9: Thank you very much for your comments. We have added to the manuscript a figure of various cellular defense mechanisms of ferroptosis and a figure of the role of these mechanisms in AD and PD. The current compounds targeting this mechanism in the treatment of PD and AD were summarized in the discussion. The content is as follows:

fig1

Figure 1. Mechanisms of ferroptosis cellular defense systems. Shows the four cellular defense systems of ferroptosis. GR, glutathione reductase; GSH, glutathione; GSSG, glutathione disulfide; GPX4, glutathione peroxidase 4; CoQH2, dihydroubiquione; CoQ, coenzyme Q; FSP1, ferroptosis suppressant protein 1; DHODH, dihydrowhey dehydrogenase; GCH1, GTP cyclohydrolase 1; DHFR, dihydrofolate reductase; BH4, tetrahydrobiopterin; BH2, dihydrobiopterin; PL-OO, phospholipid hydroperoxide; PL-OH, phospholipid alcohols; NADPH, nicotinamide adenine dinucleotide phosphate. (modified from [27,49])

Reference:

27.Wang Y, Wu S, Li Q, Sun H, Wang H (2023). Pharmacological Inhibition of Ferroptosis as a Therapeutic Target for Neurodegenerative Diseases and Strokes. Adv Sci (Weinh):e2300325.

48.Lei G, Zhuang L, Gan B (2022). Targeting ferroptosis as a vulnerability in cancer. Nat Rev Cancer, 22:381-396.

fig2

Figure 2. Mechanisms of ferroptosis cellular defense systems in PD and AD. GSH, glutathione; GPX4, glutathione peroxidase 4; CoQ10, coenzyme Q10; FSP1, ferroptosis suppressant protein 1; PL-OO, phospholipid hydroperoxide; PL-OH, phospholipid alcohols; APP, amyloid precursor protein. (modified from [100-102])

Reference:

[100]     Costa I, Barbosa DJ, Benfeito S, Silva V, Chavarria D, Borges F, et al. (2023). Molecular mechanisms of ferroptosis and their involvement in brain diseases. Pharmacol Ther, 244:108373.

[101]     Mahoney-Sanchez L, Bouchaoui H, Ayton S, Devos D, Duce JA, Devedjian JC (2021). Ferroptosis and its potential role in the physiopathology of Parkinson's Disease. Prog Neurobiol, 196:101890.

[102]     Lane DJR, Metselaar B, Greenough M, Bush AI, Ayton SJ (2021). Ferroptosis and NRF2: an emerging battlefield in the neurodegeneration of Alzheimer's disease. Essays Biochem, 65:925-940.

Table 2. Emerging compounds targeting cellular defense system for ferroptosis to attenuate PD (ACSL4, long-chain acyl-CoA synthetase 4; DFO, deferoxamine; DMT1, iron importer divalent metal transporter 1; FPN1, iron efflux transporter; FSP1, ferroptosis suppressor protein 1; FTH1, ferritin heavy chain 1; NQO1, NAD(P)H dehydrogenase[quinone]-1).

table2

Table 3. Emerging compounds targeting cellular defense system for ferroptosis to attenuate AD (ACSL4, long-chain acyl-CoA synthetase 4; DMT1, iron importer divalent metal transporter 1; FTH1, ferritin heavy chain 1; TFR1, transferrin receptor protein 1; TSG, Tetrahydroxy stilbene glycoside; γ-GC, γ-glutamylcysteine.)

table3

Line 391, “Although a growing number of studies confirm that ferroptosis plays an important role in PD and AD, the deeper molecular mechanisms are unknown, and studies with clinical translatability are still being explored. The ferroptosis cell defense system may be a promising target for the treatment of PD and AD. Since the discovery that ferroptosis is involved in the pathogenesis of neurological disorders, an increasing number of investigators have proposed targeting anti-ferroptosis to treat these diseases. An increasing number of drugs have been found to exert their therapeutic efficacy through the ferroptosis cell defense system (Table 2 3). This provides more possibilities for discovering potential therapeutic agents and therapeutic targets for PD and AD, as well as helping to further explain the pathogenesis of PD and AD. The vast majority of existing studies have been conducted in animal or cellular models, with few clinical trials. Therefore, further studies in humans through randomized trial designs are necessary to draw definitive conclusions.”

Point 10: However, Some sentences are long and confusing and should be rephrased. For example Iines 44-49 "In addition, in these neurological disorders, ferritin deposition has been shown to correlate with the pathophysiology of the disease, and particular attention has been paid to therapeutic approaches for the treatment and prevention of organ damage through pharmacological inhibition of ferritin deposition, which has opened up new opportunities for the treatment of these disorders using pharmacological inhibition of ferritin deposition [11, 15-18]."

Response 10: We would like to thank the reviewers for their positive comments on the manuscript. We have revised and edited the manuscript in English.

Reviewer 3 Report

The review article (IJMS-2570946) by Chu et al. focuses on the role of four cellular defence systems relevant in the prevention of ferroptosis.

The authors provide a clear introduction into the four defence systems and on the respective molecular mechanisms involved.

The explicit focus of the review however is on the role of ferroptosis and the individual defence systems in PD and in AD. The information provided represents a solid basis for the reader in order to get a first impression on the most relevant work in the literature. Unfortunately, the authors provide no information, concept, or hypothesis above this point.

Relevant questions with respect to the problem addressed by the title of the article e.g. could be:

- what are the differences in ferroptotic cell death in AD and in PD ?

There should be a strong focus on information that is actually available, but also on current data gaps.

- both ferroptosis and apoptosis are characterized by elevated ROS formation and oxidative stress. The apparent question is what distinguishes the two scenarios (ROS species; intracellular localization of ROS formation; levels of redox-active iron of the labile iron pool; intracelluar sequestration of the LIP (cytosol, mitochondria, lysosomes, nuclei).

Here, again, focus could be on the information that is available and on data gaps in order to direct the attention of other researchers on these open questions.

There are several comprehensive review articles on ferroptosis in PD and in AD published.

On that basis, the manuscript in its present form provides readers no significant added value in their understanding of ferroptosis in AD and PD.

For these reasons, I can not endorse acceptance of the manuscript.

The text is clearly structured and written. I would however suggest thorough proofreading by a professional language editing service.

Author Response

Response to Reviewer’s Comment

First of all, we would like to thank the reviewers for their invaluable suggestions and positive comments on the manuscript. We carefully examined these comments and have revised the manuscript point-by-point along the lines suggested.

The corrections in the revised manuscript are marked using red markers.

Reviewer 3

Point 1: There should be a strong focus on information that is actually available, but also on current data gaps. Here, again, focus could be on the information that is available and on data gaps in order to direct the attention of other researchers on these open questions.

Response 1: Thank you very much for your comments. In the discussion section, we supplemented the current research gaps in the mechanism of action of ferroptosis defense system on PD and AD according to the current research status, and summarized the progress in its treatment. As follows:

Table 2. Emerging compounds targeting cellular defense system for ferroptosis to attenuate PD (ACSL4, long-chain acyl-CoA synthetase 4; DFO, deferoxamine; DMT1, iron importer divalent metal transporter 1; FPN1, iron efflux transporter; FSP1, ferroptosis suppressor protein 1; FTH1, ferritin heavy chain 1; NQO1, NAD(P)H dehydrogenase[quinone]-1).

table2Table 3. Emerging compounds targeting cellular defense system for ferroptosis to attenuate AD (ACSL4, long-chain acyl-CoA synthetase 4; DMT1, iron importer divalent metal transporter 1; FTH1, ferritin heavy chain 1; TFR1, transferrin receptor protein 1; TSG, Tetrahydroxy stilbene glycoside; γ-GC, γ-glutamylcysteine.)

table3

Line 391, “Although a growing number of studies confirm that ferroptosis plays an important role in PD and AD, the deeper molecular mechanisms are unknown, and studies with clinical translatability are still being explored. The ferroptosis cell defense system may be a promising target for the treatment of PD and AD. Since the discovery that ferroptosis is involved in the pathogenesis of neurological disorders, an increasing number of investigators have proposed targeting anti-ferroptosis to treat these diseases. An increasing number of drugs have been found to exert their therapeutic efficacy through the ferroptosis cell defense system (Table 2 3). This provides more possibilities for discovering potential therapeutic agents and therapeutic targets for PD and AD, as well as helping to further explain the pathogenesis of PD and AD. The vast majority of existing studies have been conducted in animal or cellular models, with few clinical trials. Therefore, further studies in humans through randomized trial designs are necessary to draw definitive conclusions.”

Point 2: The text is clearly structured and written. I would however suggest thorough proofreading by a professional language editing service.

Response 2: We would like to thank the reviewers for their positive comments on the manuscript. We have revised and edited the manuscript in English.

Round 2

Reviewer 1 Report

I evaluate that the revised manuscript is very much improved, but I still have several comments and questions regarding the newly added parts.

Table 1 

1) In ‘Biochemical features’ of Ferroptosis, some words following ‘increased’ should be missing.

2) In ‘Predisposing factor’, I do not think that necroptosis is passively triggered. It is promoted by highly regulated manners.

3) Autophagy, a cellular catabolic process, is not a term for a cell death mode. Instead, ‘Autophagic cell death’, which is mediated by autophagy, should be included in this list.

Figure 2 

The conversion of PL-OO to PL-OH under GPX4 is depicted only in the AD pathway, and CoQ10 under FSP1 is depicted only in the PD pathway. Why are they differently depicted between the AD and PD pathways? 

Line 297-298, 334-335

Is it known how appoferritin up-regulates FSP1 and how Aβ decreases GPX? It should be informative to describe the link between typical features of PD and AD and the defense systems against ferrptosis. 

Author Response

Response to Reviewer’s Comment

First of all, we would like to thank the reviewers for their invaluable suggestions and positive comments on the manuscript. We carefully examined these comments and have revised the manuscript point-by-point along the lines suggested.

The corrections in the revised manuscript are marked using blue markers.

Point 1:

Table 1

1) In ‘Biochemical features’ of Ferroptosis, some words following ‘increased’ should be missing.

2) In ‘Predisposing factor’, I do not think that necroptosis is passively triggered. It is promoted by highly regulated manners.

3) Autophagy, a cellular catabolic process, is not a term for a cell death mode. Instead, ‘Autophagic cell death’, which is mediated by autophagy, should be included in this list.

Response 1: Thank you very much for your suggestion. We have made the following changes:

 table1

Point 2:

Figure 2

The conversion of PL-OO to PL-OH under GPX4 is depicted only in the AD pathway, and CoQ10 under FSP1 is depicted only in the PD pathway. Why are they differently depicted between the AD and PD pathways?

Response 2: Thank you very much for your suggestion. We apologize for the neglect of detail that appears in the figure and have made the following changes:

 fig2

Point 3: Line 297-298, 334-335

Is it known how appoferritin up-regulates FSP1 and how Aβ decreases GPX? It should be informative to describe the link between typical features of PD and AD and the defense systems against ferrptosis. 

Response 3: We have adapted and added to the description of the problem. The content is as follows:

Line 295-302, The pathological feature of PD is the damage of dopamine neurons in the substantia nigra pars compacta. This study indicates that reduced FSP1 may lead to an oxidized CoQ10 state, which further exacerbates lipid peroxidation, leading to the development of ferroptosis and ultimately the degeneration of DA neurons in PD. However, apoferritin could effectively inhibit the production of lipid peroxidation through FSP1 upregulation, thereby effectively suppressing ferroptosis and preventing the loss of dopamine neurons in the substantia nigra pars compacta, and exerting its neuroprotective effect on PD.

Line 337-341, The histopathology of AD is characterized by extracellular aggregation of Aβ plaques and intracellular aggregation of neuroprogenitor fiber tangles. Aβ1-40 (subtypes of Aβ) causes Aβ to cross the blood-brain barrier into the blood by inducing mitochondrial autophagy and inhibiting GPX4 from inducing pericytes to undergo ferroptosis, leading to the development of AD.

Reviewer 3 Report

-

Author Response

We would like to thank the reviewers for their invaluable suggestions and comments on the manuscript.